# Dysregulation of miRNAs in DLBCL: Causative Factor for Pathogenesis, Diagnosis and Prognosis

**DOI:** 10.3390/diagnostics11101739

**Published:** 2021-09-22

**Authors:** Mohammed Alsaadi, Muhammad Yasir Khan, Mahmood Hassan Dalhat, Salem Bahashwan, Muhammad Uzair Khan, Abdulgader Albar, Hussein Almehdar, Ishtiaq Qadri

**Affiliations:** 1Department of Biological Science, Faculty of Science, King AbdulAziz University, Jeddah 21589, Saudi Arabia; maalsaadi1@kau.edu.sa (M.A.); khattak.yasir1@gmail.com (M.Y.K.); abdulkade_albar@hotmail.com (A.A.); hmehdar@kau.edu.sa (H.A.); 2Hematology Research Unit, King Fahad Medical Research Center, King AbdulAziz University, Jeddah 21589, Saudi Arabia; smbahashwan1@kau.edu.sa; 3Vaccine and Immunotherapy Unit, King Fahad Medical Research Center, King AbdulAziz University, Jeddah 21589, Saudi Arabia; 4Department of Biochemistry, Faculty of Science, King AbdulAziz University, Jeddah 21589, Saudi Arabia; mahmoodalhat@gmail.com; 5Department of Hematology, Faculty of Medicine, King AbdulAziz University, Jeddah 21589, Saudi Arabia; 6King AbdulAziz University Hospital, King AbdulAziz University, Jeddah 21589, Saudi Arabia; 7Department of Health Sciences, City University of Science and Information Technology, Peshawar 25000, Pakistan; muhammad.uzair@cusit.edu.pk; 8Department of Microbiology, Faculty of Medicine, Jeddah University, Jeddah 23218, Saudi Arabia

**Keywords:** DLBCL, miRNAs, genetics, epigenetic, prognosis, diagnosis

## Abstract

MicroRNA is a small non-coding RNA (sncRNA) involved in gene silencing and regulating post-transcriptional gene expression. miRNAs play an essential role in the pathogenesis of numerous diseases, including diabetes, cardiovascular diseases, viral diseases and cancer. Diffuse large B-cell lymphoma (DLBCL) is an aggressive non-Hodgkin’s lymphoma (NHL), arising from different stages of B-cell differentiation whose pathogenesis involves miRNAs. Various viral and non-viral vectors are used as a delivery vehicle for introducing specific miRNA inside the cell. Adenoviruses are linear, double-stranded DNA viruses with 35 kb genome size and are extensively used in gene therapy. Meanwhile, Adeno-associated viruses accommodate up to 4.8 kb foreign genetic material and are favorable for transferring miRNA due to small size of miRNA. The genetic material is integrated into the DNA of the host cell by retroviruses so that only dividing cells are infected and stable expression of miRNA is achieved. Over the years, remarkable progress was made to understand DLBCL biology using advanced genomics and epigenomics technologies enabling oncologists to uncover multiple genetic mutations in DLBCL patients. These genetic mutations are involved in epigenetic modification, ability to escape immunosurveillance, impaired *BCL6* and NF-κβ signaling pathways and blocking terminal differentiation. These pathways have since been identified and used as therapeutic targets for the treatment of DLBCL. Recently miRNAs were also identified to act either as oncogenes or tumor suppressors in DLBCL pathology by altering the expression levels of some of the known DLBCL related oncogenes. i.e., miR-155, miR-17-92 and miR-21 act as oncogenes by altering the expression levels of *MYC*, SHIP and FOXO1, respectively, conversely; miR-34a, mir-144 and miR-181a act as tumor suppressors by altering the expression levels of SIRT1, *BCL6* and CARD11, respectively. Hundreds of miRNAs have already been identified as biomarkers in the prognosis and diagnosis of DLBCL because of their significant roles in DLBCL pathogenesis. In conclusion, miRNAs in addition to their role as biomarkers of prognosis and diagnosis could also serve as potential therapeutic targets for treating DLBCL.

## 1. Introduction

MicroRNA is about 22 nucleotides small non-coding RNA (sncRNA) molecule that functions as a regulator of post-transcriptional gene expression and RNA silencing [1]. In 1998, Andrew Fire and Craig C. Mello discovered the interference RNA(RNAi) in animals [2], and were awarded Nobel prize in physiology or medicine in 2006 [3]. Two main types of sncRNAs, i.e., siRNA and miRNA account for the interference mechanism of RNA [4]. siRNA are small RNA molecules with approximately 25 nucleotides possessing both sense and anti-sense polarity [5]. Primarily it was discovered in plants that were undergoing virus induced gene silencing or co-suppression as compared to control plants. Later on, it was also detected in Drosophila tissue culture by induction of long exogenous dsRNA and, also, from extracts of Drosophila embryo [6]. miRNAs are non-coding RNAs playing a significant role in gene expression and are transcribed from DNA sequences to precursors miRNA and lastly mature miRNAs [7]. miRNAs typically have incomplete base pairing to inhibit many RNAs while siRNA are derived from long double-strand RNA (dsRNA) precursors and base-pair perfectly to cleave mRNA specifically at a single specific site [8].

## 2. miRNA Biogenesis

The two-step miRNA synthesis is typically a nuclear and cytoplasmic breakdown process involving ribonuclease III, Drosha and Dicer [9]. miRNA synthesis in humans is initiated with RNA polymerase II transcription which binds to promoter region making a hair-pin loop of pre-miRNA. However, RNA polymerase III transcribes some miRNA that initiates primary transcripts [10]. The pri-miRNA contains a hair-pin stem, terminal loop and flanking single stranded sequence from hundred nucleotides to several kilo bases (kb) [11] and at 5 end (5′ UTR), modified nucleotide covers pri-miRNA that makes poly-adenylated (having several adenosine; poly A tail) [11,12]. Nuclear protein DiGeorge critical region 8 (DGCR 8) also known as Pasha in invertebrates and RNA-III enzyme recognizes the hair pin shaped miRNA [13]. Drosha and DGCR8 work with another complex network of proteins known as microprocessors [14]. The pri-mRNA is cleaved by microprocessor to form a shorter hairpin approximately 70 nucleotides in length [15]. The resulting product possesses two nucleotide overhangs at 3′ end (3′UTR) and 5 (5′UTR), this is referred as precursor-miRNA (pre-miRNA) to which downstream are necessary for effective processing [16]. Exportin5 protein (XPO5) is a karyopherin member that recognizes a two-nucleotide overhang at 3′ end of pre-miRNA hairpin deserted by Drosha and this XPO5 protein transports Pre-miRNA from nucleus to cytoplasm. [17]. The transfer of pre-miRNA is an energy dependent pathway using GTP attached to Ran protein [18].

Inside cytoplasm pre-miRNA is cleaved by Dicer (RNAase III enzyme) by interacting with the 5′ and 3′ end, thus removing the loop and resulting in 22 nucleotide miRNA duplex [19]. Either strand of miRNA potentially serve as functional miRNA but just one miRNA strand is merged in RNA-induced silencing complex (RISC) [20].

## 3. Delivery Platforms for miRNA

One of the important features to performing experiments with miRNA is to select a safe and efficient delivery vehicle. Nowadays, both viral and non-viral vector platforms are used for delivery of miRNA.

### 3.1. Viral Vectors

Four main types of viral vectors are used in delivering miRNA into cell i.e., Lentiviral vector, adenovirus vector, adeno-associated viral vectors (AAV) and retroviral vectors [21]. Adenoviruses are linear, double-stranded DNA viruses with 35 kb genome size and are extensively used in gene therapy [22]. These vectors are used to express different types of RNAi but the major drawback is Adenovirus associated RNA inhibits RNAi by competing with XPO5 and DICER, thus resulting is suppression of RISC activity [23,24]. Meanwhile, AAVs are more frequently used for miRNA gene therapy as compared to adenoviral vectors. AAVs are single stranded DNA viruses that accommodate up to 4.8 kb foreign genetic material and are favorable for transferring miRNA due to small size of miRNA gene [25]. AAVs are promising gene delivery tools due to their lack of ability to facilitate long-term episomal expression, and minimal oncogenic potential and inflammatory response [26]. Currently many clinical trials are underway utilizing AAV platform for gene therapy. AAVs are commonly used in preclinical studies for cancers, muscular Dystrophies, neurodegenerative disorders and viral infections [26,27].

The tremendous ability to infect post mitotic cells and efficient transduction in neurons make it a promising vector. Pre-existing adaptive and innate immune responses in hosts are main challenges and barriers in AAV gene transfer. To date, 258 studies have been registered in Clinicaltrials.gov using AAV but none have reported encoding miRNA or siRNA and to be administered in humans. Viral delivery systems a remarkably efficient system, however, these may elicit immunogenic responses which halt their effectiveness [26].

Retroviruses are RNA viruses capable of carrying an 8 kb foreign genome and integrate into the host cell genome [28]. The genetic material is integrated into the DNA of the host cell so that only dividing cells are infected and stable expression of miRNA is achieved [29]. Lentiviruses are a subgroup of retroviruses with common features of integration in the host genome but can infect both dividing and non-dividing cells [30]. Due to this ability, they infect terminal differentiated cells to treat neurologic diseases [31].

### 3.2. Non-Viral Vectors

This delivery approach is less immunogenic, less toxic and has a minimum limitation in terms of size of transferred gene, but it has poor transfection efficiency. The most common is lipid-based approach termed as liposomes, lipid molecules with encapsulated nucleic acids. Liposomes have a shorter life span in vivo due to non-specific binding of proteins in serum [32]. The utilization of polymers i.e., poly lactide-co-glycolide (PLGA), polyethyleneimine (PEI) and cell penetrating peptides (CPP) are reported to be effective in term of less toxicity to membrane and cells [33]. Contrary to usage of polymers, a number of studies have reported usage of nanoparticles [30], the most frequently used nanoparticles are Fe3O4 based nanoparticles and gold nanoparticles (AuNPs) [34].

The oncogenic and tumor suppressing abilities of several miRNA make it promising a candidate for RNA based therapeutics. The only siRNA based FDA-approved drug Patisiran was licensed in 2018 for treating a rare polyneuropathy caused by transthyretin-mediated amyloidosis [35].

No miRNA-based drugs have come to clinic, but many are under clinical trials. MRX34 is liposome encapsulated double stranded RNA drug that mimics (miR-34) that is human tumor suppressor. MRX34 was first miRNA-based drug to enter human clinical trial and showed tumor suppression in 29% of patients [36]. However, FDA stop phase 1b due to severe immune related toxicity in five patients. Further studies proved that lipid encapsulation is not the cause of severe immune toxicity [37,38]. Cobomarsen a locked nucleic acid-based oligonucleotide inhibitor of miRNA-155 was studied to check safety and efficacy in clinical trial phase 1. Result showed reduced lesion burdens and acceptable safety profile in patients with hematological cancer. These results prompt the initiation of clinical trial 2 of cobomarsen versus vorinostat, an FDA-approved drug for CTCL [39]. Thus, non-viral vector platform for delivering miRNAs is accepted extensively due to low immunogenicity and safety profile [39].

## 4. Diffuse Large B-Cell Lymphoma

Diffuse Large B-cell Lymphoma (DLBCL) is an aggressive cancer of mature B lymphocytes and the most common subtype of non-Hodgkin’s lymphoma, it comprises about 40% of all non-Hodgkin’s lymphoma cases. It has a highly variable clinical course; although most of affected patients initially show a good response to chemotherapy, approximately half of them will achieve a durable remission [40].

DLBCL is genetically heterogeneous disease, which can be divided into ABC-DLBCL and GCB-DLBCL subtypes at different levels of gene expressions, while subgroups are based on the B-cell markers and NF-κB action pathways [41,42].

Heterogeneity in DLBCL is attributed to chromosomal translocation and single nucleotide variation. i.e., mutation in exon 15 of *EZH2* results in gain of function of *EZH2* implicating suppression of major DLBCL regulator genes [43,44]. The orientation of DLBCLs in specific topographic locality includes CNS and primary cutaneous DLBCL having specific gene expression GEP (genomic profile) different from the nodal DLBCL [44].

## 5. Etiology of DLBCL

Many chromosomal transitions result in the production of mutated proteins, resulting in the gain or loss of function protein. Mutated proteins downregulate specific pathways which prevent cells from normal cell death/ apoptosis and favor cell growth [45,46]. Some major factors such as immunosuppression and some class of infectious viruses such as human herpesvirus-8, human herpesvirus-6, Epstein-bar virus (EBV) and hepatitis-C virus are associated with human B-cell lymphomas [47,48]. Cancer cell types adopt a pathological mechanism to lower the chances of recognition by NK cells resulting in their prolonged life. A significant role of PDL-1 was found in oncogenic cells which promote immune evasion [49,50]. Genome profiling studies showed high alteration in histone and chromatin enzymes contributing to lymphomagenesis. DLBCL pathogenesis resulted from the alteration of proto-oncogene expression and suppressor genes caused by the accumulation of genetic lesions [51]. Earlier studies only focused on single nucleotide mutation which was later addressed by Bjoern Chap et al. 2018 finding significant driver genes and five new remarkable subsets which could be further used for combinational therapies [52].

## 6. Epigenetic Modifications in DLBCL

Findings in genome of DLBCL patients have revealed high mutational frequency affecting epigenetic machinery; the most reported epigenetic modifications are histone acetyltransferases (HAT) and histone methyltransferases [53]. In HAT, mutations cause loss of function (LoF) in *CREBBP* and *EP300*, detected in about 35% of DLBCL patients with significantly high mutations seen in GCB-subtype [54,55]. The *CREBBP*/*EP300* protein adds acetyl groups to lysine residues on histones and non-histone nuclear proteins, acting as transcriptional coactivators for several transcription factors that bind to DNA [56].

Consistently, *CREBBP*/*EP300* activates transcription through different epigenetic mechanisms, comprising of targeted chromatin acetylation and transcriptional activators such as p53 and *BCL6* acetylation-mediated inactivation [56,57,58]. Mutation in the *CREBBP* could cause C-terminal truncation in the HAT domain or change the amino acid residue(s), which weakens its capacity of acetylating *BCL6* and p53 which are its substrates, in turn resulting in the activation of oncoprotein and reduced tumor suppressive role of p53 for the regulation of the DNA damage response reaction which occurs in the germinal center during immunoglobulin genes remodeling [59]. The consequence of *BCL6* action suppressing p53 activity could lead to an amplified DNA damage tolerance by deregulation of the apoptotic and cell cycle response arrest [59,60]. The involvement of HATs in chromatin remodeling and transcriptional regulation suggests the consequences of impaired function of **CREBBP*/*EP300** due to alterations. Further studies on the role of **CREBBP*/*EP300** deregulation would help in the mechanistic understanding of all the targets that significantly affect DLBCL pathogenesis [61].

Histone methyltransferases such as MLL2 (mixed lineage leukemia 2) are among the identified candidates for DLBCL pathogenesis whose mutations account for over 30% of DLBCL [52,62]. The MLL2 gene is highly conserved and ubiquitously expresses the MLL2 protein. MLL2 protein regulates gene transcription by adding a methyl group to the 4th position of lysine residue in histone 3 domain (H3K4) [63]. The H3K4 trimethylation is a preserved mark of chromatin that is transcriptionally active and narrowly related to the promoters of active genes and counteracts the transcriptional repression executed by the methylation of H3K9 and H3K27 [63]. Mutation of MLL2 in DLBCL generates a truncated protein that lacks the catalytic SET domain, which is essential for the methyltransferase activity, known to be associated with tumor suppressor activity. It is important to note that *MLL3,* the paralogue of MLL2 was also found to contain an indel (insertion or deletion) mutation which accounts for approximately 15% of DLBCLs [64,65].

*EZH2* encodes for a trimethylated H3K27 histone methyltransferase which causes repressed gene expression [66]. A gain of function (GoF) mutation in *EZH2* detected in about 22% of GCB-DLBCL patients is correlated with poor prognosis [67]. The *EZH2* GoF implicates in amplified H3K27me3 stage through transformed substrate selectivity due to preserved tyrosine residue (Tyr641) within the domain of *EZH2* SET [62]. *EZH2* is a promising therapeutic target for DLBCL treatment, as several studies involving *EZH2* mutated DLBCL cells have shown that small molecule inhibitors targeting *EZH2* could induce apoptosis and cell cycle arrest [68,69]. Taken together, mutational changes in chromatin remodeling genes could have a significant effect on transcriptional regulation and favors the lymphomagenesis through epigenetically remodeling lymphoma cells [70].

## 7. Immune Escape Pathways in DLBCL

Genetic mutations involved in immune recognition and antigen-presenting functions account for significant mutational frequencies seen in both molecular subtypes of DLBCL. The disruption of beta-2 microglobulin gene is reported in 30% of DLBCL patients which is caused due to the Biallelic deletion and inactivating mutations [71]. The invariant region of the major histocompatibility complex (MHC) class 1 is encoded by B2M gene, found on the surface of nucleated cells. Approximately 75% of DLBCL patients lack expressed B2M gene [71]. B2M is essential for human leukocyte antigen (HLA) development and identification of cytotoxic lymphocytes (CTL) [71]. Involvement of other epigenetic mechanisms was suggested as fractions of DLBCL patients were observed to have reduced expression of B2M but possess elevated HLA-I expression [72]. Truncating mutations and focal deletion were detected in cluster of differentiation 58 (CD58) locus in 20% of DLBCL subjects. [73,74]. CD58 is an immunoglobulin ligand of CD2 expressed on surfaces of T-cells and natural killer cells (NK) that participate in the activation of these cells [74]. The absence of B2M, HLA-I and CD58 in 60% of DLBCL patients signifies the role of lymphomagenesis in protecting cell-mediated lysis of CTL and NK cell [70]. Other mutations disrupting the immune response regulators comprise of alteration in different MHC class II trans-activator genes (CIITA), programmed cell death ligand (PDL2) and PDL1, all of which are preferentially detected in primary mediastinal large B cell lymphoma (PMBCL) [74]. The downregulation of HLA-II reorganizations of CIITA may decrease tumor cells immunogenicity. In contrast, the amplification of PDL1 is shown to be associated with impaired anti-tumor immune responses in different cancer types [75]. These gene mutations are essential in immune response regulated by tumor microenvironment (TME) activities which aid tumor cells to escape immune surveillance mechanisms [76].

## 8. Alterations of NF-κβ Activity and BCR Signaling in DLBCL

Gene expression profiling has revealed the role of NF-κB in DLBCL poor prognosis and disease progression, as it was observed that the DLBCL expression profiles are enriched with NF-κB targeted genes resulting in cell survival and cell proliferation of DLBCL cells [77]. Many researchers have shown through molecular profiling that constitutive activation of of NF-κB is associated with interleukin-1 receptor (1L-1 R), Toll-like receptor (TLR) and B-cell receptor (BCR) signaling pathways [78]. In BCR signaling pathway, the existence and development of B-cell lymphomas are associated with mutations in *CD79A* and *CD79B*. The *CD79A* and *CD79B* mutations account for >20% of overall mutations in ABC-DLBCL [79]. In addition, the loss of function in BCR negative regulators (i.e., DGKZ, SLA MAP4K1, LYN, PTPN, LAPTM5 and PRKCD) result in prolonged BCR signaling [80]. Of note, knockdown of proximal and distal sub-units associated with BCR, of specifically toxic to ABC-DLBCL, contributing to potential to the development of BCR-targeted therapies for ABC-DLBCL subtype [80]. The use of ibrutinib, a BTK inhibitor, was shown to be effective in treating patients with ABC-DLBCL having *CD79A/B* alterations. Similarly, a small molecule inhibitor of mucosa-associated lymphoid tissue translocation protein 1 (MALT1) showed a selective action against cell lines of ABC-DLBCL in xeno-transplanted tumor and in vitro studies [81].

The alterations in oncogenic myeloid differentiation primary response protein 88 (*MYD88*) were found in about 30% of patients with ABC-DLBCL [82]. The mutation in *MYD88* involves the substitution of leucine with proline at position 265 (L265P), leading to GoF, thereby triggering NF-κβ and JAK/STAT3 transcriptional responses by interacting with lkB kinase (comprising lkBkB and lkBkG) [82].

About 60% of alterations in both negative and positive regulators of NF-κβ were reported in cases of ABC-DLBCL while a lesser percentage were reported in cases of GCB-DLBCL [78]. These mutational changes inactivate certain genes, encoding for tumor-necrosis factor alpha-induced protein 3/A20 (TNFAlP3/A20), which are predominant in ABC-DLBCL [83]. TNFAlP3/A20 is associated with inactivation of NF-κβ, BCR and TLR responses. Moreover, TNFAlP3/A20 inactivates BCR response through its interaction with MALT1 [81]. Therefore, LoF of TNFAlP3/A20 could influence lymphomagenesis by inducing prolonged NF-κβ signaling. Approximately 9% of ABC-DLBCLs patients were shown to possess CARD11 mutations [84]. CARD11 is an essential component of CBM (CARD11/BCL10/MALT1) complex, which alongside TRAF6 is essential for the transduction of signaling from BCR to NF-κβ [84]. While an expected outcome of these genetic mutations initiates responses of NF-κβ signaling pathway similar signals could activate several additional downstream signaling cascades such as PI3K, and ERK/MAP kinase. In summary, BCR and NF-κβ pathways are disrupted by genetic mutations in both ABC-DLBCL and GCB-DLBCL.

## 9. Impaired *BCL6* Activity in DLBCL

*BCL6* is a transcription factor controlled by *CREBBP*/*EP300*, which binds to conserved sequence in the promoter region of targeted genes, thus controlling gene expression through interaction with specific co-repressor complexes [85]. *BCL6* is associated with biological activities of DLBCL cells by suppressing the expression of wide range of genetic factor included in numerous signaling pathways including CD40 in BCR signaling, T-cell mediated B-cells inactivation, the ability to detect as well as respond to DNA damage through repressing of the TP53 protein in TP53/ATR mediated DNA damage response, the induction of apoptosis by suppressing BCL2 in BCL2/BCL2L1 mediated apoptosis and plasma cells differentiation through *PRDM1*/BLIMP1 suppression [85]. *BCL6* associated activities sustain the proliferation of GCB cells while undergoing tolerance for DNA damage and breaks, connected with class switch recombination and somatic hyper-mutation (SHM) without triggering the DNA damage responses. Furthermore, *BCL6* causes premature B-cell inactivation and promotes its exit from the germinal centers (GC) before the clone survival selection thereby producing antibodies attracted to antigens [57]. *BCL6* chromosomal mutation is observed in 35% of both ABC and GCB DLBCL, with higher mutational frequencies in the ABC-DLBCL subtype [53]. Recombination events in DLBCL cells affect the downstream of integral *BCL6* coding domain to heterological sequences, formed by more than 20 different chromosome partners [86]. Moreover, downregulation in BCL-6 expression is correlated with the differentiation of post-GC B-cells, achieved by chromosomal translocation of *BCL6* [87]. Less frequently, *BCL6* associated mutations prevent IRF4-mediated *BCL6* repression, related to CD40/CD40L interaction in light zone GC [85]. *BCL6* activity is also impaired by another set of genetic mutations affecting the post-transcriptional regulatory mechanism of *CREBBP*/*EP300*, which impairs the acetylation-mediated *BCL6* inactivation and *FBXO11* mutations that occurs in 5% of DLBCL cases [88]. The *FBXO11* mutation is associated with *BCL6* proteosomal degradation via SRP1/CUL1/SCF complex. *MEF2B* is a transcription factor expressed in GC, and also acts as *BCL6* transcription regulator. The GoF mutation of *MEF2B* is seen in approximately 10–18% DLBCL patients; mostly predominant in GCB-subtype [89]. In lymphomagenesis initiation, *BCL6* play a critical role confirmed using in vivo mouse model and the results demonstrated that deregulation of *BCL6* could promote the increase of human-like DLBCL [69,90]. Connecting all evidence of the critical role of *BCL6* in lymphomagenesis, the *BCL6* can be used as a promising target for therapeutic intervention [90]. Of note, studies have shown that *BCL6* inhibitors have a potential anti-lymphoma capacity and have a strong synergistic effect in DLBCL combinational treatments [90].

## 10. Blocking the Terminal Differentiation Pathway in DLBCL

The alteration in the terminal differentiation pathway is one of the predominant events that occur in DLBCL. Downregulation of *BCL6* is essential to reduce the expression of the terminal differentiation gene targets, including plasma cell master regulator *PRDM1*/BLIMP1 [91]. The differentiation regulatory axis is altered in majority of ABC-DLBCL cases because of mutually exclusive mutations leading to LoF in BLIMP1. The LoF in BLIMP1 could directly cause deletions and disruptive mutations usually observed in 25% of DLBCL patients or indirectly by *BCL6* deregulation through suppression seen in 25% of DLBCL patients [92]. Alternatively, the LoF of *PRDM1* and its interaction to *BCL6* chromosomal translocation may represent another oncogenic mechanism affecting similar pathway to influence lymphomagenesis by hindering lethal variations. Consistent with the BLIMP1/PRDMl/*BCL6* mechanisms, deletion in BLIMP1 causes proliferative lymphoma disease similar to that of human ABC-DLBCL [92].

## 11. Role of miRNA in DLBCL Pathogenesis

Many aberrantly expressed miRNAs are identified as critical pathogenic factors in DLBCL. It is important to note that many miRNAs are involved in regulating the development of other hematological cancers. These miRNAs have an essential role in the development of normal B-cells in healthy humans. In DLBCL, miR-17-92 is over-expressed, suggesting its role as potential oncogene as shown in vivo using DLBCL induced mice [93].

Remarkably, it was observed that mice overexpressing miR-17-92 developed lymphoproliferation and increased *MYC* expression. However, lymphocytes from miR-17-92 knockdown mice exhibited increased expression of pro-apoptotic genes, such as *PTEN* and *BIM* [94]. Functional genomic studies have shown the transactivation of *MYC* through miR-17-92 cluster, both of which result in DLBCL progression by silencing tumor suppressor genes [95].

The regulatory axis of miR-17-92 comprises of *MYC* and its role in the *MYC*/miR-17-92/E2F circuit resulting in upregulation of miR-17-92 cluster that targets the cell proliferation regulator; *E2F1* (Figure 1). Consistently, the supporting proliferation regulator gene *E2F3* assists *E2F1* in promoting the expression of miR-17-92 cluster. Recently, *PTPROT* and *PP2A* were reported to target miR-17-92 clusters and, also, activate the BCR-signaling in DLBCL cells [96]. Furthermore, miR-217 was also reported to stabilize the *BCL6* expression as a regulator of germinal center formation. Conversely, another study reported that miR-144 act as tumor suppressor by suppressing the *BCL6* function in DLBCL xenografted mice [97]. Both miR-10a and miR-187 act as tumor suppressors by regulating *BCL6* expression through induction of cell apoptosis as shown in cell culture and tumor microarray studies suggesting these miRNAs could be exploited as a novel therapeutic target for DLBCL treatment [98,99].

Another miRNA; miRNA-222-3p, was shown to have an oncogenic role in DLBCL by promoting cell proliferation, invasion and inhibiting apoptosis by silencing phosphatase2 regulatory subunit B alpha (*PPP2R2A*) in DLBCL. Research evidence has shown that miRNA-222-3p promotes downregulation of *PPP2R2A* and tumor growth in xenografted mice [100].

In cancer, particularly in B-cell lymphoproliferative disease, miR-155 is reported to act as oncogene (oncomiR) miRNA. However, in normal conditions miR-155 plays significant role in the activation and maturation of B-cell lymphocyte. Several studies have attributed miR-155 overexpression to several lymphomas, including DLBCL. miR-155 mediated targeting of SH2 domain which contains *C/EBPb* and inositol-5-phosphatase-1 (*SHIP-1*) may increase the accumulation of malignant pre-B cells in mice models due to interaction with miR-155 [101]. In xeno-transplant DLBCL cell model, TNF-α cell growth targeting *SHIP1* is regulated by miR-155 overexpression. The miR-155 mediated *SHIP2* regulation is demonstrated by the suppressive effect of miR-155 on growth-inhibitory factors such as *TGF-β1* and *BMP2/4* in DLBCL cell through *SMAD5* suppression by targeting *PIK3R1* and regulating PI3K-AKT pathways [96]. Moreover, miR-155 also target *TGFβR2* suggesting a novel oncogenic pathway contributing to the development of tumors. miR-155 is shown to target HGAL, a GC marker and lymphocyte mobility inhibitors and NF-κβ promote the expression of miR-155 [102]

Another study has reported miR-181a as a negative regulator of NF-κβ signaling [103]. Overexpression of miR-181a significantly decreased the expression and activity of crucial NF-κβ signaling components in DLBCL, leading to reduced tumor cell proliferation and survival by targeting CARD11, *NFKB1A*, *NFKB1*, *RELA* and *REL* [103]. The expression of miR-181a was shown to be negatively correlated with CARD11; a master regulator of DLBCL [103].

Overexpression of miR-34a is associated with poor prognosis and positive feedback loop in DLBCL. The tumor suppressor, *p53*; induces the expression of miR-34a which silences a NAD deacetylase called *SIRT1* whose effect enhances and stabilizes *p53* mediated activators. In DLBCL, high-grade transformation is promoted by low expression of miR-34a through dysregulation of *FOXP1* expression which is inhibited by *MYC* [104]. Downregulation of miR-34a and miR-21 up-regulation was reported to be linked with relapse-free survival. miR-21 acts as an oncogene by directly targeting the tumor suppressors *PTEN*, *PDCD4* and *FOXO1*, which leads to the activation of PI3K/AKT/mTOR oncogenic pathway. Study involving miR-21 inhibition illustrated the significant decrease in the DLBCL cell lines and sensitivity to R-CHOP treatments and reduction in invasion and proliferation of tumor cell [104].

Recently, studies reported the role of miRNAs as tumor suppressors by inhibiting *PD-L1* and long non-coding RNAs (lncRNAs) such as *MALAT1* and *NEAT1*. A study using patients’ tissues and cell culture has shown that downregulation of miR-214 is correlated with *PD-L1* upregulation in DLBCL compared with normal control showing that both genes have antagonistic effect [105]. Therefore, overexpression of miR-214 is associated with reduction in cell invasion, cell viability and increased apoptosis [105]. Similarly, *PD-L1* and *MALAT1* were reported to have antagonistic effect with miR-195, showing the role of miR-195 as tumor suppressor by decreasing expression level of both *PD-L1* and *MALAT1*. This interaction showed that overexpressed miR-195 correlates with cell proliferation, cell migration, immune escape and increased apoptosis in DLBCL [106]. Another miRNA; miR-34b-5p, was shown to silence NEAT1 and GLI1 in the tissues and cell lines of DLBCL by attaching to 3′ UTR mRNA transcripts *NEAT1* and *GLI1* thereby preventing their expression [107].

In addition, miR-26a was reported to regulate *CDK5* activity and *p53* expression demonstrating its importance in cell survival, cell proliferation and cell cycle progression. The role of miR-26a in regulating both *p53* and *CDK5* makes it a suitable therapeutic target for DLBCL treatment [108]. Similarly, upregulation of *CDK6* due to downregulation of miRNA-320d was reported in DLBCL [109]. Overexpression of *CDK6* is related to poor prognosis in DLBCL patients. miR-320d act as tumor suppressor by silencing the *CDK6* mRNA transcripts at 3′ UTR resulting in inhibition of cell proliferation and cell cycle arrest. miRNA-320d is a promising therapeutic target due to the tumor suppressive function [109].

Overexpression of miR-23a was reported to suppress metastasis suppressor 1 (*MTSS1*) in DLBCL. miR-26a is negatively correlated with *MTSS1* expression. MTSS1 functions as a suppressor of cell proliferation, tumor metastasis and, also, play a key role in actin cytoskeleton recombination [110]. It was suggested that miR-23a promotes the proliferation, invasion and metastasis in DLBCL by silencing *MTSS1* [110]. Similarly, miR-645 targets the *DACH1* by binding to 3′-UTR of *DACH1* mRNA transcript leading to reduced *DACH1* expression in DLBCL cells. Reduced expression of *DACH1* is associated with cell proliferation and survival. *DACH1* is expressed in different cancer types, and it is correlated with poor prognosis and tumor progression of cancer patients. Specifically, in DLBCL, decreased *DACH1* is inversely correlated with increased miR-645 level. Overexpressed miR-645 is associated with cell proliferation and apoptosis [111].

miR-101 acts as tumor suppressor by targeting the 3′ UTR of *KDM1A,* thereby reducing tumor growth and progression. In addition, the *KDM1A* mediated suppression by miR-101 suggests its impact in regulating apoptosis and cell proliferation through MAPK/Erk signaling pathway [112].

Aberrant expression of miR-4638-5p was reported to be associated with high *ERG* expression. Compared with ERG-negative; the ERG-positive DLBCL is more likely to harbor mutations in genes that are essential in cell cycle control, BCR mediated signaling and β-catenin degradation [113]. Functional and clinicopathological studies associated with ERG-related miRNAs (i.e., miR-4638-5p) and pathways could provide new information into the pathogenesis of DLBCL and reveal novel targets for the management of patients with DLBCL [113]. Similarly, miR-520c-3p acts as tumor suppressor by targeting *eIF4GII* in xenografted mice [114]. Overexpressed *eIF4GII* in DLBCL is associated with abnormal protein synthesis, resulting in increased cell proliferation. Thus, the results of in vivo and in vitro studies confirm miR-520c-3p level is correlated with the expression of *eIF4GII* in DLBCL [114].

## 12. The Role of miRNAs in Diagnosis and Prognosis in DLBCL

MiRNAs are shown to regulate cancer progression in DLBCL by acting either as oncogenes or tumor suppressors. MiRNAs influence the two major DLBCL molecular subtypes, GCB-DLBCL and ABC-DLBCL. GCB-DLBCL and ABC-DLBCL subtypes are distinguished based on miRNA profiling [115,116,117,118]. Several studies reported the role of miRNAs in DLBCL diagnosis, thereby distinguishing it from other types of lymphomas such as follicular lymphoma (FL) and Burkitt lymphoma (BL) [119,120].

Relapse in central nervous system (CNS) associated with DLBCL, occur in 5% patients with low survival rate [121]. DLBCL patients with CNS relapse tend to have high expression of miR-30d and miR-20a. Therefore, miR-30d and miR-20a expression profiles could be used to stratify DLBCL patients [121]. Similarly, miR-155 is also being linked to prognosis, metastasis and treatment failure in DLBCL patients. A study using mouse model showed overexpression of miR-155 is associated with a high-grade lymphoma development [122].

Conversely, a complete recovery was observed in mice when miR-155 stimulus was removed in an inducible expression system [123]. Moreover, the low expression of both miR-34a and miR-27b are associated with poor prognosis in DLBCL patients [123]. Similarly, shorter relapse-free survival in DLBCL is linked with low miR-21 expression in serum and tissue samples. As a result, miR-21 level is proposed to function as an independent prognostic biomarker for DLBCL [124,125]. In DLBCL patients, the presence of miR-21 is suggested to contribute to increased cellular viability and escape apoptotic in tumor cells by targeting *BCL2* and *PTEN* in apoptotic and cell proliferation pathways. MiR-21 inhibition is reported to increase DLBCL cell lines sensitivity to the R-CHOP regimen (hydroxydaunorubicin, rituximab, oncovin, prednisone, cyclophosphamide regimen), which causes decreased tumor cell proliferation and invasion [126]. Transcriptome-wide association analysis revealed the association between aberrant miRNA expression profile with prognostic result in DLBCL patients treated with R-CHOP regimen [127]. The aberrantly expressed miRNAs included miR-330, miR-199b, miR-27a, miR-519, miR-222, miR-425 and miR-142 associated with overall survival [124], whereas miR-130a and miR-125b were linked with R-CHOP resistance [128]. Overexpression of miR-497 and miR-199a were reported to be associated with increased sensitivity to doxorubicin, rituximab, and vincristine drugs present in the R-CHOP regimen. Similarly, increased sensitivity to doxorubicin and rituximab was attributed to overexpression of miR-409-3p, miR-370-3p and miR-381-3p [129].

In the serum samples of DLBCL patients, the levels of miR-155, miR-210 and miR-21were significantly expressed as compared to healthy controls [130]. Another study using plasma also recorded high expression of miR-124 and miR-532-5p and low expression of miR-424, miR-345, miR-145, miR-122, miR-425, miR-128, miR-197 and miR-141 [131].

In serum samples of DLBCL patients, high expression of miR-155, miR-16, miR-29c, miR-15 and low expression of miR-34a was reported [132]. In DLBCL, a high expression level of serum miR-22 was associated with poor prognostic results [132]. Recently using NGS technology, 51 differentially expressed miRNAs were identified in DLBCL patient samples [133]. Three miRNAs were confirmed by quantitative reverse transcriptase-polymerase chain reaction (qRT-PCR), showing that miR-431-5p and miR-323-3p were downregulated and miR-34a-5p was upregulated (Figure 2) [133].

## 13. Treatment

Based on the newest classification by World Health Organization (WHO), more than 100 different types of lymphoma, most of which are B-cell lymphomas and different diagnosis, clinical characteristics and treatment choices has been reported [134]. Due to heterogeneity of DLBCL, the prophecy and choice of treatment strategies are difficult [135]. Globally, immunochemotherapy with doxorubicin, cyclophosphamide, prednisone, (R-CHOP) rituximab and vincristine is considered as the current first line treatment [136]. The introduction of rituximab, a chimeric monoclonal antibody immunoglobulin surface CD20, the R-CHOP treatment has become the current standard of treatment of choice for DLBCL irrespective of the disease subtype [126]. However, after primary response, up to 40% of patients experience relapses or early treatment failure with this treatment approach [137].

A study by Lawrie et al. using microarray expression reported different miRNAs that are linked with prognosis of DLBCL patients treated by R CHOP, including miR-519, miR-222, miR-27a, miR-425, miR-199b, miR-330, miR-302 and miR-142 [94].

A similar study reported CHOP treatment of miRNAs with five different levels provided significant results with miRNAs: miR-33a, miR-455-3p, miR 520d-3p, miR-1236 and miR-224 [126]. However, in DLBCL cases, chemoresistance association resulted from miR-130a and miR-125b was found [122]. MiR-34a may be used as a new therapeutic target of DLBCL [133]. Cytotoxic effect of CHOP regimen significantly increased when the cells of CRL2631 were transfected by miR-21 with anti-sense oligonucleotides. By using quantitative reverse transcriptase polymerase chain reaction (qRT-PCR), miR-21 significantly increased DLBCL cell line sensitivity to CHOP treatment and cause tumor cell proliferation reduction [138]. miR-21 inhibition leads to reduced tumor cell proliferation and invasion in DLBCL [125]. The over expression of miR-187 induces cell apoptosis in vitro highlighting its role as therapeutic target of DLBCL [98].

Although there is currently no FDA-approved miRNA-based drug as medical intervention for DLBCL, efforts were made to target some of the important miRNAs known to mediate DLBCL pathogenesis, as shown in clinical trials. For MRX34; a miR-34a mimic and cobomarsen (MRG-106), anti miR-155 drugs were shown to possess anti-tumor activities in clinical trials involving cancer patients including DLBCL patients [36,139].

MRX34, a mimetic of a miR-34a was used in clinical trials for the treatment of cancer patients. Although it showed promising outcomes as an anti-tumor miRNA-based drug, but it was stopped by the FDA due to its immunotoxic effect on cancer patients [36].

Cobomarsen targets and negatively regulates miR-155 expression in DLBCL, which in turn prevents resistance to growth inhibitory factors; TGFβ1 and TGFβ2 and inhibit *MYC* stability thereby having overall effect on DLBCL cell proliferation. Clinical trial (NCT02580552) studies have reported the safety and efficacy of cobomarsen in DLBCL patients [39].

## 14. Conclusions

Deep insight into different miRNAs and its expression could be a powerful tool in diagnosing, differentiating between different subtypes of non-Hodgkins lymphoma, and treating DLBCL. Currently, most published data regarding miRNAs and treating DLBCL are carried out in vitro studies. More studies should be conducted on transgenic organisms carrying human genes for better implication and understanding therapeutic roles of miRNAs in DLBCL. Conventional drugs with combination of different miRNAs could be a potential drug of choice for treating DLBCL in future after passing successful clinical trials.

## Figures and Tables

**Figure 1 diagnostics-11-01739-f001:**
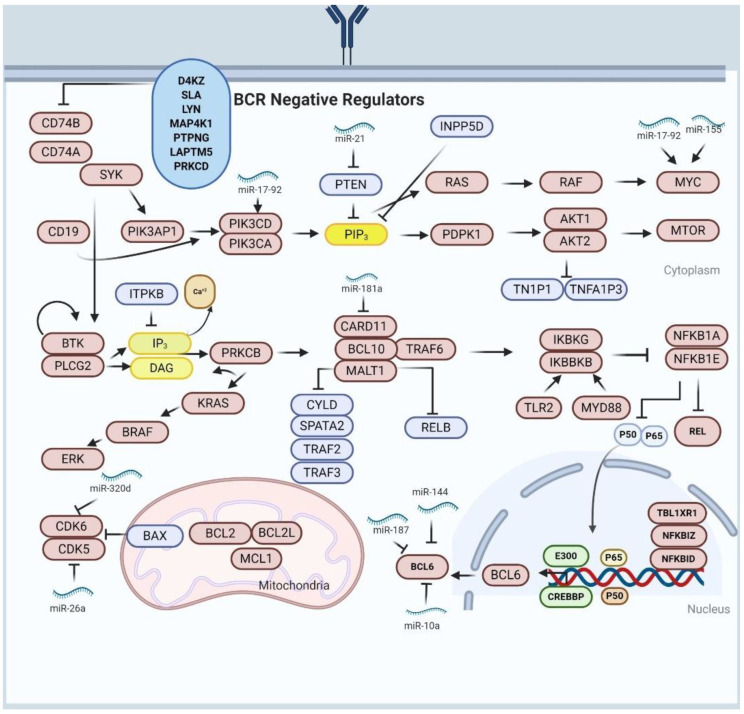
The association of miRNAs and altered signaling pathways in DLBCL pathogenesis. The brown colored proteins are the oncogenes, and the blue colored proteins are tumor suppressors. The oncogenic miRNAs promote upregulation of oncogene or inhibit tumor suppressor activity, conversely, the tumor suppressive miRNAs promote upregulation of tumor suppressor or inhibit oncogenes activity. Both oncogenic and tumor suppressive miRNAs are used as biomarkers for prognosis and diagnosis of DLBCL.

**Figure 2 diagnostics-11-01739-f002:**
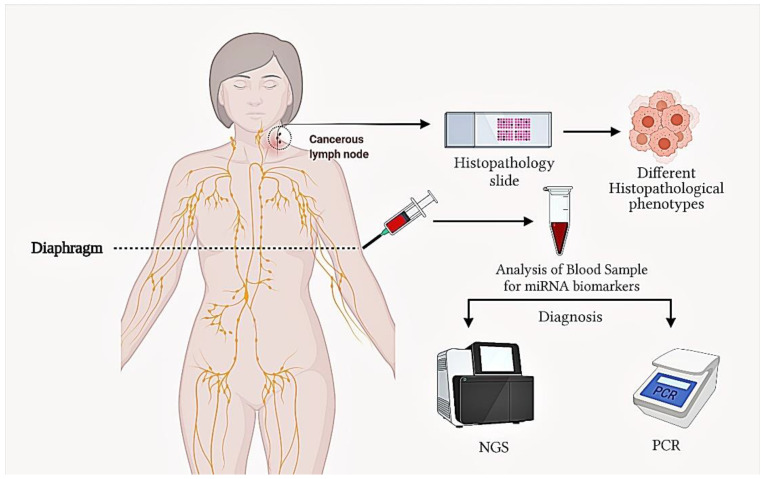
Cancerous lymph node and blood diagnosis for DLBCL by histopathological procedures, PCR and next generation sequencing (NGS).

## Data Availability

Not applicable.

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
