# Peer review of "Dysregulation of miRNAs in DLBCL: Causative Factor for Pathogenesis, Diagnosis and Prognosis"

_diagnostics, 2021, doi:10.3390/diagnostics11101739_

Round 1

Reviewer 1 Report

The manuscript presents a nice overview of the known possible roles of miRNAs in DLBCL pathogenesis. However, complex processes, leading to tumor formation, progression and drug resistance is likely influenced by many other factors. It would be useful to at least mention other types of RNA molecules, including lncRNAs. This would broaden the scope of the manuscript and increase its contribution to global understanding of DLBCL pathophysiology.

Reviewer 2 Report

Major concerns:

  1. Much of the literature cited is >10-15 years old. While certainly important to acknowledge seminal studies in a field, the authors need to focus their attention on more recent works that reflect the current methodologies and that describe ongoing challenges for miRNA-based therapeutics used in the clinical setting.
  2. Section 3: Delivery platforms are described. These all work in vitro for expressing and/or delivering exogenous miRNAs into cells. However, how many of these platforms are currently used in vivo? Used in patients? Are there examples of AAV-siRNAs or AAV-miRNAs given to patients? Please describe current methodologies for miRNA therapies (in cancer, for example). Please describe the key limitations/challenges for these types of strategies and well as miRNA delivery platforms that are (i) utilized in the clinic, (ii) in clinical trials, and (iii) under active development and headed into clinical trials.
  3. Section 12 and 13: There are several miRNA-based treatments for B cell NHL (and other cancers) already in phase I clinical trials. Why are none of these mentioned?? miR-34 and miR-155 are linked to poor prognosis for DLBCL. Are there therapeutics against these miRNAs? How might miRNA profiling be used to evaluate drug responses for DLBCL? Lines 426-444: For this section, perhaps describe first (i) miRNAs that are differentially regulated in circulation/tissues in DLBCL patients versus healthy individuals and second (ii) the miRNAs that alter sensitivity to therapy or can be used as biomarkers for drug resistance.
  4. Sections need to be edited substantially for proper English language and grammar. The manuscript currently reads like a rough draft. There are a few lines highlighted below that do not make sense as currently worded, plus many more that need to be revised.

Other comments:

Section 2 (lines 59-82): This section describes miRNA biogenesis steps, but no function, so why is it titled “miRNA Biogenesis and function”? Also, miRNA biogenesis has been heavily reviewed by experts in the field—the authors should consider shortening Section 2 and referring to recent literature/reviews rather than citing papers that are 15-20 years old. Figure 1 is not really necessary for the topic of this review.

Sections 6-10: These sections start to get slightly off-topic. How do epigenetic changes in DLBCL impact miRNA expression and function? How are immune escape pathways and NFkB/BCR signaling pathways modulated by miRNAs in DLBCL?

A figure should be included for section 11 that illustrates the miRNA interactions with specific targets and shows how these interactions impact oncogenic/tumor suppressor signaling pathways.

Figure 2: Perhaps include tissue specimens (i.e. lymph node biopsies) that are collected for miRNA analysis/DLBCL prognosis.

Line 45: Mello and Fire described the *mechanism* of RNA-interference in animal cells.

Line 126: “in exon 15 of gene EZH2 resulting in the gain of function”. Do the authors mean mutations occur in exon 15 of EZH2? Please revise.

Line 138: A reference is needed for NK cells.

Line 491: This sentence does not make sense: “Inspire of vast scientific studies conducted every year on the role of miRNA in its urge of time to explore new targets for DLBCL treatment.” Please revise.

Round 2

Reviewer 2 Report

Figure 1 is a nice addition.

Major points to include for discussion have been adequately addressed, and modest improvements have been made to the text to make it more readable.

The manuscript needs another round of editing for English language/grammar/spell-check as there are still words that are not correctly spelled and sentences that are incoherent.

Here is just one of many examples: line 108: "Viral dilevery sytems a remarkbly efficent system but due to eliciting immunogenic responses halt its effectivness"
I think this is supposed to read: "Viral delivery systems are remarkably efficient systems; however, these may elicit immunogenic responses which halt their effectiveness."
